# Increase in Autoantibodies-Abzymes with Peroxidase and Oxidoreductase Activities in Experimental Autoimmune Encephalomyelitis Mice during the Development of EAE Pathology

**DOI:** 10.3390/molecules26072077

**Published:** 2021-04-04

**Authors:** Anna S. Tolmacheva, Kseniya S. Aulova, Andrey E. Urusov, Irina A. Orlovskaya, Georgy A. Nevinsky

**Affiliations:** 1Institute of Chemical Biology and Fundamental Medicine, SB of the Russian Academy of Sciences, 630090 Novosibirsk, Russia; anny_@mail.ru (A.S.T.); amaya.rain.nsu@gmail.com (K.S.A.); urusow.andrew@yandex.ru (A.E.U.); 2Institute of Clinical Immunology, Siberian Branch of the Russian Academy of Medical Sciences, 630090 Novosibirsk, Russia; irorl@mail.ru

**Keywords:** EAE model, C57BL/6 mice, catalytic antibodies, peroxidase and oxidoreductase activities

## Abstract

The exact mechanisms of multiple sclerosis (MS) development are still unknown, but the development of experimental autoimmune encephalomyelitis (EAE) in C57BL/6 mice is associated with the violation of bone marrow hematopoietic stem cells (HSCs) differentiation profiles associated with the production of harmful for human’s autoantibodies hydrolyzing myelin basic protein, myelin oligodendrocyte glycoprotein (MOG_35–55_), and DNA. It was shown that IgGs from the sera of healthy humans and autoimmune patients oxidize many different compounds due to their H_2_O_2_-dependent peroxidase and oxidoreductase activity in the absence of H_2_O_2_. Here we first analyzed the change in the relative redox activities of IgGs antibodies from the blood of C57BL/6 mice over time at different stages of the EAE development. It was shown that the peroxidase activity of mice IgGs in the oxidation of ABTS (2,2′-azino-bis(3-ethylbenzothiazoline-6-sulfonic acid) is on average 6.9-fold higher than the oxidoreductase activity. The peroxidase activity of IgGs increased during the spontaneous development of EAE during 40 days, 1.4-fold. After EAE development acceleration due to mice immunization with MOG_35–55_ (5.3-fold), complexes of bovine DNA with methylated bovine serum albumin (DNA-metBSA; 3.5-fold), or with histones (2.6-fold), the activity was increased much faster. The increase in peroxidase activity after mice immunization with MOG_35–55_ and DNA-metBSA up to 40 days of experiments was relatively gradual, while for DNA-histones complex was observed its sharp increase at the acute phase of EAE (14–20 days). All data show that IgGs’ redox activities can play an important role in the protection of mice from toxic compounds and oxidative stress.

## 1. Introduction

Multiple sclerosis (MS) is known as a chronic demyelinating disease of the central nervous system (CNS). Its etiology remains unclear, and the most widely accepted theory of MS pathogenesis assigns a leading role to the destruction of myelin by the inflammation related to autoimmune reactions [1]. There are several different experimental autoimmune encephalomyelitis (EAE) models, including C57BL/6 mice, mimicking a specific facet of human MS (for a review, see [2,3,4]). EAE in C57BL/6 mice has a spontaneous chronic-progressive course. Autoimmune diseases (AIDs) were supposed first to be originated from hematopoietic stem cells (HSCs) defects [5]. The spontaneous and antigen-induced development of systemic lupus erythematosus (SLE) in MRL-lpr/lpr mice [6,7,8] and EAE in C57BL/6 mice [9,10,11] was later demonstrated is a consequence of the immune system-specific reorganization of bone marrow HSCs. The immune system defects in AIDs consist of specific changes in the profile of bone marrow HSCs differentiation combined with the production of specific catalytic abzymes-auto-antibodies (Abs) hydrolyzing DNAs, RNAs, polysaccharides, peptides, and proteins [6,7,8,9,10,11].

Antibodies-abzymes against transition states of different chemical reactions catalyzing more than 150 various reactions are novel biologic catalysts (for review see [12,13,14,15]). Natural catalytic abzymes slipping DNA, RNA, polysaccharides, oligopeptides, and proteins were revealed in the sera of patients with many autoimmune diseases (for review, see [16,17,18,19,20]). With some exceptions, abzymes with these enzymatic activities in healthy volunteers are absent or usually extremely low. The auto-Abs with catalytic activities was revealed as statistically significant and the earliest markers of many autoimmune diseases’ development, which appear immediately after the start of a change in the differentiation profile of bone marrow stem cells [15,16,17,18,19,20]. Therefore, the early stages of development of AIDs can be detected only by changes in these parameters [9,10,11,12,13,14,15,16,17,18,19,20].

DNase abzymes of SLE [21] and MS patients [22] are harmful since they are cytotoxic and induce cell apoptosis; they may play a very negative role in the pathogenesis of both pathologies. In MS and SLE patients, abzymes against myelin basic protein (MBP) and myelin oligodendrocyte glycoprotein (MOG_35–55,_ further designated as MOG) with proteolytic activity may attack MBP of the myelin-proteolipid sheath of axons. Consequently, these auto-Abs may also play an essential harmful role in MS pathogenesis.

The partially reduced oxygen species in all higher organisms (O_2_^•^, H_2_O_2_, and OH^−^) are produced as intermediates of aerobic respiration as well as appear in organisms through exposure to ionizing radiation. They act as potent oxidants attacking different cellular DNA, various proteins, lipids, and etc. [23,24,25]. Oxidative damages of different cell compounds were regarded as significant factors in mutagenesis, carcinogenesis, and aging. Several canonical antioxidant enzymes (catalases, peroxidases, superoxide dismutases, and glutathione peroxidases) provide critical defense mechanisms for preventing cell and blood components from oxidative modifications [25,26,27]. Mammalian antioxidant enzymes are present mostly in different cells, but their low enzymatic activities can be revealed in the blood since they lose their enzymatic activities in the blood relatively fast [28,29]. In contrast to classical antioxidant enzymes, antibodies exist in the blood for a long time (1–3 months) [30].

Therefore it was exciting to determine whether blood catalytic abzymes can participate in the protection of humans and animals from oxidative stress. It was first shown that IgGs from blood sera of healthy Wistar rats in the presence (H_2_O_2_-dependent peroxidase activity) and in the absence of H_2_O_2_ (peroxide-independent oxidoreductase activity) similarly to horseradish peroxidase could oxidize effectively 3,3′-diaminobenzidine (DAB) and several other compounds [31,32,33,34,35]. Besides, IgGs of healthy Wistar rats demonstrated superoxide dismutase and catalase activities [36]. Using different methods, we have shown convincing evidence that H_2_O_2_-dependent peroxidase and H_2_O_2_-independent oxidoreductase activities are intrinsic to IgGs of healthy humans and animals [31,32,33,34,35,36,37,38]. We have then compared redox activities of IgGs from sera of healthy humans and patients with SLE and MS [39]. Average peroxidase and oxidoreductase activities of SLE IgGs were statistically significantly higher in comparison with abzymes from healthy humans. IgGs of MS patients demonstrated these activities more elevated than that for healthy volunteers but lower than for SLE abzymes [39]. The data obtained speak in favor that abzymes can serve as an additional blood factor of reactive oxygen species detoxification. In addition, the protection of SLE and MS patients from some harmful compounds may be somewhat better than healthy peoples [39].

It was interesting how classical enzymes and antibodies with redox activities are similar or different. Several canonical antioxidant enzymes (catalases, peroxidases, and superoxide dismutases) are metal dependent enzymes, while glutathione peroxidases are selenium-dependent enzymes (not metal ions-dependent) [28]. Peroxidase and oxidoreductase activities of rat IgGs are Me^2+^-dependent [31,32,33]. However, during standard methods of antibodies purification, they only partially lose metal ions and exhibit a high level of catalytic activity without the addition of exogenous metal ions [32]. Complete suppression of the activity of rat antibodies is observed only after the addition to the reaction mixtures of EDTA at high concentration, 0.1–0.3 M [32]. A completely different situation was found in the case of IgGs from the blood sera of humans and mice [37]. These polyclonal antibodies, in contrast to those in rat mice, exhibited both metal-dependent and metal-independent redox activities. It was shown that after a standard procedure of these IgGs purification, they contain several metal ions, which relative amount in overage decrease in the order: Fe ≥ Pb ≥ Zn ≥ Cu ≥ Al ≥ Ca ≥ Ni ≥ Mn >> Co ≥ Mg [37]. After dialysis of IgGs against EDTA or adding EDTA (0.1–0.3 M) to the reaction mixtures, their peroxidase and oxidoreductase activities decreased by only 50–54% [37]. The question was why human IgGs catalyze oxidation of substrates in the absence of metal ions similar to selenium-dependent glutathione peroxidase. Finally, using synchrotron radiation X-ray fluorescence method, it was shown that human IgGs contain 0.4 ± 0.1 μg selenium/1 g of IgGs [37]. Thus, it was shown that human IgGs containing selenocysteine can catalyze metal-independent oxidation of substrates.

It was interesting to understand at what stages of developing the autoimmune diseases the activation of the synthesis of abzymes with peroxidase and oxidoreductase activities can occur. C57BL/6 mice were recently used to study possible mechanisms of spontaneous, MOG- and polymeric bovine DNA-accelerated development of EAE [9,10,11]. It was shown that the main immunogen stimulating the development of SLE and several other AIDs is not DNA but its complex with histones [40]. Moreover, high-affinity anti-DNA Abs have been revealed as a major component of intrathecal IgGs in the brain and cerebrospinal fluid (CSF) cells of MS patients [41]. Therefore, earlier in studies of the accelerated development of EAE in C57BL/6 mice, two different DNA complexes were used: polymeric bovine DNA-histones and DNA-methylated bovine serum albumin (metBSA) [10,11]. Positively-charged metBSA similar to positively-charged histones forms strong complexes with DNA and therefore simulates well DNA-histones complexes in the production of anti-DNA antibodies [10].

In addition, one of the best accelerators of EAE development in C57BL/6 mice is MOG_35–55_ [2,3,4,9]. Spontaneous development of EAE over time occurs relatively slowly and smoothly demonstrating gradual increase in the catalytic activities of the abzymes [9,10,11]. The mice treatment with DNA-histones, DNA-metBSA, and MOG results in powerful acceleration of EAE development associated with very strong increase in the enzymatic activity of antibodies from 6–7 to 20 days (onset and acute phases of the pathology) [9,10,11]. Despite very similar patterns of EAE development, nevertheless, in the case of each of these immunogens, noticeable differences were also observed [9,10,11].

Therefore, we used a mouse model of MS (EAE C57BL/6 mice), in which case it is possible to analyze changes in the activity of antibodies during the onset, acute phase, and remission of this pathology. Besides, we compared previously obtained data on specific overtime changes in the profile of bone marrow HSCs differentiation and appearance in the blood of catalytic abzymes splitting DNA, MOG, and MBP with in time changes in IgGs peroxidase and oxidoreductase activities.

## 2. Results

### 2.1. Experimental Groups of Mice

As mentioned above, the development of EAE leads to the immune system of C57BL/6 mice specific reorganization associated with significant changes in differentiation of mice HSCs, the increase in proliferation of lymphocytes in different organs, and significant suppression of cell apoptosis in these organs [9,10,11]. All these defects-changes in the immune system lead to increased proteinuria, the generation of catalytically active Abs hydrolyzing DNA, MBP, and MOG [9,10,11]. As noted above, the development of autoimmune (AI) diseases in humans leads to an increase in the concentration of antibodies-abzymes with redox enzymatic activities [38,39]. It was of interest to see how these defects-changes in the immune system can affect possible alterations in antibodies’ relative activities in the oxidation of various substrates. Besides, it seemed important to compare the overtime patterns of changes in the relative activity of Abs with oxidative functions with those for abzymes that hydrolyze MBP, MOG, and DNA during mice EAE development [9,10,11].

In the study of oxidoreductase activities, we have used homogeneous IgGs preparations containing no any canonical enzymes before and after immunization of C57BL/6 mice with MOG [9], DNA-metBSA [10], and DNA-histones [11]. To demonstrate the processes of violation of the immune status in C57BL/6 mice, Appendix A indicate over time changes in a number of mice bone marrow BFU-E [erythroid burst-forming unit (early erythroid colonies)], CFU-E [erythroid burst-forming unit (late erythroid colonies)], CFU-GM [granulocyte-macrophage colony-forming unit], and CFU-GEMM [granulocyte-erythroid-megacaryocyte-macrophage] colony forming units (Appendix A), the relative amount of lymphocytes in bone marrow, spleen, thymus, and lymph nodes (Appendix A), and level of cell apoptosis (%) in different organs (Appendix A) for untreated mice, as well as after their treatment with DNA-histones, DNA-metBSA, and MOG described in [9,10,11].

### 2.2. Criteria Analysis of Catalytic Activities of Antibodies

As previously was shown, the optimal substrates for IgG antibodies with peroxidase activity from the blood of healthy rats [31,32,33,34,35,36], healthy donors, and patients with AIDs are 2,2′-azino-bis(3-ethylbenzothiazoline-6-sulfonic acid (ABTS), 3,3′-diaminobenzidine (DAB), and *o*-phenylenediamine (OPD) [37,38,39]. To prove that mouse IgGs’ peroxidase activity is their intrinsic property and is not due to co-purifying enzymes, we have applied strict criteria. IgG_mix_ (IgG_mix_ preparations; mixtures of 21 IgG preparations after mice treatment with MOG [9], DNA-metBSA [10], and DNA-histones [11]) were electrophoretically homogeneous (Figure 1, lane C).

To exclude possible hypothetical traces of contaminating peroxidases, three IgG_mix_ preparations were subjected to SDS-PAGE. The peroxidase activity was detected after gel incubation in the reaction mixture containing DAB. Yellow-brown bands were revealed only in the positions of three intact IgG_mix_ (Figure 1, lanes 1–3). Since SDS dissociates all protein complexes, revealing the activity only in the gel region corresponding to three intact IgG_mix_ preparations and the absence of any other protein bands and the activity provides direct evidence all IgG_mix_ preparations possess intrinsic peroxidase activity.

### 2.3. Substrates of IgGs

As previously was shown, the optimal substrates for IgG antibodies with peroxidase activity from the blood of healthy rats [31,32,33,34,35,36], healthy donors, and patients with AIDs are ABTS, DAB, and OPD [37,38,39]. It was necessary to choose which of these substrates is best suited for analyzing a large number of IgG preparations from the blood plasma of mice. To evaluate the substrate providing the maximum activity of the mouse IgG preparations, mixtures of several preparations (IgG_mix_ preparations) corresponding to the treatment of mice with three antigens (MOG [9], DNA-metBSA [10], and DNA-histones [11]) were used. It has been shown that all three substrates are efficiently oxidized by antibodies from the blood of mice immunized MOG, DNA-metBSA, and DNA-histones. As an example, Figure 2 shows four typical kinetic curves of ABTS, DAB, and OPD oxidation (in their optimal concentrations [37,38,39]) by these IgG_mix_ preparations in the presence (peroxidase activity) and in the absence of hydrogen peroxide (oxidoreductase activity).

The relative apparent *k*_cat_ values (*k*_cat_ = V (M/min)/[IgG] (M)) characterizing oxidation by IgG_mix_ preparations of ABTS, DAB, and OPD in their fixed optimal concentrations [37,38,39]) were assessed. It can be seen that the oxidoreductase activity, according to apparent *k*_cat_ values in the case of any of the three substrates, is 1.2–6.9-fold lower than the peroxidase activity. In addition, ABTS is the best substrate for mice IgGs with peroxidase activity, the oxidation of which occurs 5.7–9.0 and 211 times faster than DAB and OPD, respectively. Taking this into account, the relative activities of various IgG preparations corresponding to different stages of EAE development in mice were evaluated using ABTS as substrate. Unfortunately, the amount of IgGs for a complete analysis of oxidoreductase activity changes at different stages of EAE development before and after mice immunization with various antigens was not enough.

### 2.4. In Time Changes in the Peroxidase Activity during the Development of EAE

To evaluate a possible change in the relative activity of individual IgG in time before and after mice immunization with various antigens, their relative peroxidase activities in the oxidation of ABTS were determined. For further analysis, the averaged data of apparent *k*_cat_ values (seven mice in each group) were used. Figure 3a demonstrated the changes in average *k*_cat_ values of IgGs corresponding to spontaneous development of EAE. For comparison, the data on the same IgG preparations’ relative activity in the hydrolysis of DNA and MOG received in [9,10,11] are presented. It can be seen that at 3 months of age, the blood of C57BL/6 mice contains IgGs active in the oxidation of ABTS. The spontaneous development of EAE during 63 days leads to a relatively gradual increase in peroxidase activity by a factor ~1.7, which is nearly parallel with the rise in the activity of IgGs in the hydrolysis of DNA and MOG.

As one can see from Appendix A, significant changes in the differentiation profile of HSCs after C57BL/6 mice treatment with MOG begin at 6–7 days with a peak of changes at 18–20 days after the immunization. According to [2,3,4], 6–7 days after the immunization correspond to the onset, while 18–20 days to the acute stages of EAE disease development, the remission phase begins after about 25–30 days. At 6–7 days, when in parallel with the change in the HSCs differentiation profile, it is observed a significant increase in the MOG-, and DNA-hydrolyzing activity of IgGs, which by 20 days increase by approximately 3.3 and 10 times, respectively (Figure 3b). During the period of remission, there is a sharp decrease in these activities. The time dependences of the relative IgG-dependent hydrolysis of MOG and MBP after mice immunization with MOG, DNA-metBSA, and DNA-histones are very similar [9,10,11] and therefore they are not shown in the Figures of this article.

It is interesting that, in contrast to these DNase and MOG-hydrolyzing activities, the relative efficiency of ABTS oxidation increases almost gradually to 40 days 5.3-fold (Figure 3b). At the same time, during the spontaneous development of EAE, IgG’s peroxidase activity by 40 days increases only ~1.4 times. In other words, immunization of mice with MOG leads to an acceleration of the production of abzymes with peroxidase activity by a factor of 3.8. Nevertheless, there is no correlation between the dependence of increase in the peroxidase activity with profiles of changes in HSCs differentiation and the synthesis of abzymes that hydrolyze DNA and MBP (Figure 3b).

Not only immunization of mice with MOG [9], but also with DNA-metBSA [10] and DNA-histones [11] accelerate the development of EAE. At the same time, the patterns of bone marrow HSCs differentiation profiles after immunization of mice with MOG and DNA-metBSA differ significantly (Appendix A). This leads to different changes in the dependences of antibodies’ activities in the hydrolysis of DNA and MBP after mice immunization with DNA-metBSA compared to those after immunization with MOG (Figure 3b,c). In contrast to the immunization of mice with MOG, there is no significant increase in DNA-hydrolyzing activity at 7–20 days after immunization with DNA-metBSA. There is a considerable delay in producing abzymes with DNase activity after immunization with DNA-metBSA, which begins to increase only after 25 days, but by 40 days, it increases by ~122 times (Figure 3c). A slight delay is observed in the production of abzymes against MOG, and it also begins to overgrow only after 25 days. By 20 days after immunization of mice with MOG, the MOG-hydrolyzing activity of antibodies increases 5.5 times, while with DNA-metBSA only 2-fold (Figure 3b,c).

Interestingly, after mice treatment with DNA-metBSA, IgGs’ peroxidase activity increases almost gradually up to 20 days, and then it begins to increase faster. The peroxidase activity change profile during 40 days does not correlate with those for the DNase and MOG-hydrolyzing activities of antibodies (Figure 3c). By the fortieth day after mice immunization with MOG, the peroxidase activity increases 5.3-fold, and in the case of DNA-metBSA only 3.5 times. Consequently, MOG better stimulates the production of abzymes with peroxidase activity than DNA-metBSA by a factor of 1.5.

Differentiation profiles of bone marrow HSCs after immunization of mice with DNA-histones differ from those for MOG and DNA-metBSA (Appendix A). As in the case of mice immunization with MOG, after their treatment with DNA-histones, there is no noticeable increase in the DNase activity of IgGs at 7–20 days, but similar to the treatment with DNA-metBSA a significant increase in this activity is observed after 25 days (Figure 3d). The growth of MOG-hydrolyzing activity after mice treatment with DNA-histones, as in the case of their immunization with MOG, begins from 6–7 days, but in the period from 15 to 40 days, it reaches a plateau. It is interesting that only in the case of immunization of mice with DNA-histones, the profile of changes in the oxidative activity of IgGs over time (Figure 3d) to some extent correlates with the overtime patterns of changes in IgGs MOG-hydrolyzing activity after mice immunization with MOG (Figure 3b,d). In this case, there is a sharp increase (2.6-fold) in the peroxidase activity of IgGs in the period from 7 to 20 days, and then its slow raises to 40 days (Figure 3d).

Taken together, spontaneous development of EAE leads to a relatively gradual increase in the activity of antibodies with peroxidase activity to some extent similar to the growth in the activities of IgGs that hydrolyze DNA and MOG (Figure 3a). Immunization of mice with MOG (Figure 3b) and DNA-metBSA (Figure 3c) leads to accelerated production of abzymes oxidizing ABTS compared to spontaneous development of EAE at 40 days by factors 3.7 and 2.5, respectively. These in time dependencies in peroxidase activity increase are somewhat gradual. It is much unexpected that only after treating mice with DNA-histones, rapid growth in peroxidase activity at 7–20 days is observed (Figure 3d). However, at 40 days, this activity is higher than that after the spontaneous development of EAE only 1.5 times.

We estimated the values of *K*_m_ (3.8 × 10^−4^ M) and *k*_cat_ (386.7 min^−1^) for one of IgG preparation with the highest activity after immunization of mice with MOG in oxidation of ABTS (Figure 4).

Unfortunately, the amount of IgGs for a complete analysis of H_2_O_2_-independent oxidoreductase activity changes at different stages of EAE development before and after their treatment with different antigens was not enough. Using IgG_mix_ preparations corresponding to the beginning of the experiment (zero time) and 40 days after their immunization, it was shown that the activity of the mixtures of antibodies with oxidoreductase activity increases approximately: MOG (5.1-fold), DNA-metBSA (3.0-fold), and DNA-histones (2.3-fold). Thus, the relative activity of Abs in the oxidation of ABTS by peroxidase-IgGs is higher than oxidoreductase-abzymes but their relative changes in time before and after mice immunization with different antigens are comparable.

## 3. Discussion

It is known that reactive oxygen and nitrogen species (ROS) play an important role in the processes of oxidative stress [42]. Oxidative disorders are an integral part of the pathological processes in many AI diseases [42,43,44,45,46,47], including MS [48,49]. ROS and subsequent oxidative damages may contribute to MS’s formation and persistence by acting on particular pathological processes [50]. ROS initiates lesions by inducing blood-brain barrier disruption, reinforcing myelin phagocytosis, and leukocyte migration. They contribute to lesion persistence by mediating cellular damage and biological macromolecules important for CNS cells’ functioning [42,43,44,45,46,47,48,49,50].

Endogenous antioxidant enzymes usually counteract oxidative stress. It has been shown that in MS, they include overexpressed superoxide dismutase 1 and 2, catalase, and heme oxygenase 1 [50]. There is no doubt that, as in the case of all mammals, other antioxidant enzymes are also directed against oxidative stress. However, antioxidant enzymes are mostly present in different cells, and their enzymatic activities in the blood are low because, in the blood, they quickly lose their activities [28,29]. Stable blood molecules are immunoglobulins that remain intact for several months [30]. It was previously shown that the blood of healthy donors and rats contains antibodies-abzymes with antioxidant peroxidase and oxidoreductase activities [31,32,33,34,35,36,37,38]. In addition, it was shown that abzymes with these activities in the blood of SLE and MS patients are statistically significantly higher than in healthy donors [39]. However, to analyze these activities, IgG preparations were taken from patients’ blood at different stages of the development of these pathologies. Taking this into account, it was interesting to understand how the peroxidase and oxidoreductase activities of antibodies can be changed during different stages of AIDs development. For this purpose, we used the analysis of in time possible changes in the peroxidase activity of C57BL/6 mice antibodies during the spontaneous and antigen-induced development of EAE.

It was shown previously that spontaneous development of EAE in C57BL/6 mice is associated with a change in differentiation profile of bone marrow HSCs, cell apoptosis, and lymphocyte proliferation in several organs [9,10,11] (see Supplementary Appendix A). Immunization of C57BL/6 mice with MOG, DNA-metBSA, and DNA-histones results in parallel acceleration of EAE and production of harmful abzymes hydrolyzing DNA, MOG, and MBP [9,10,11]. Taking this into account, it was interesting to compare the time dependences of changes in the activity of polyclonal antibodies in the hydrolysis of DNA and MOG with those for IgG-dependent peroxidase and oxidoreductase activities. Previously it was shown that the peroxidase activity of IgGs from the blood of healthy donors and patients with SLE and MS is much higher than the peroxide-independent oxidoreductase activity of the same polyclonal Abs. A similar result was obtained when comparing these activities from the blood plasma of C57BL/6 mice. On average, the peroxidase activity of IgGs in the case of any of the three substrates (ABTS, DAB, and OPD) was 1.2–6.9-fold times higher than the oxidoreductase activity. Therefore, further, during the development of EAE in mice, we analyzed the change in the peroxidase activity of antibodies in the oxidation of ABTS as the best substrate.

Interestingly that the average relative activity of IgG preparations from 28 three months old mice C57BL/6 mice before their immunization in the oxidation of ABTS according to the apparent values of *k*_cat_ (69.8 ± 23.0 min^−1^) is comparable for this value for healthy humans (73.7 ± 12.0 min^−1^ [37]). Peroxidase activity after spontaneous development of EAE (63 days) by mice increased ~1.7-fold to (*k*_cat_ = 120.0 ± 20.0 min^−1^), which is also to some extent comparable with *k*_cat_ value (169.0 ± 34.0 min^−1^), characterizing peroxidase oxidation of ABTS by IgGs from MS patients [39]. The relative peroxidase activity compared to the spontaneous development of EAE during 40 days (100.0 ± 12.0 min^−1^), after mice immunization increases in the following order: DNA-histones (141.0 ± 11.7 min^−1^), DNA-met-BSA (246.0 ± 11.7 min^−1^), and MOG (372.0 ± 40.0 min^−1^) (Figure 3). Thus, the relative activities of abzymes in the oxidation of ABTS from the blood of humans and mice are, to some extent, comparable. It was interesting to compare the affinity of ABTS for abzymes of humans and mice. Taking this into account, we estimated the values of *K*_m_ (3.8 × 10^−4^ M) and *k*_cat_ (386.7 min^−1^) for one of IgG preparation with the highest activity after immunization of mice with MOG (Figure 4). The affinity for ABTS of the most active IgGs of patients with SLE ((5.0 ± 0.4) × 10^−4^ M) and MS ((4.7 ± 0.4) × 10^−4^ M) in term of *K*_m_ values were comparable with that for Abs of mice. The *k*_cat_ value for IgGs of SLE (321.0 ± 25.0 min^−1^) was to some extent comparable to that for mice antibodies, and in the case of MS patients (169.0 ± 14.0 min^−1^) it was remarkably lower [39].

As indicated above, the development of EAE and the production of various abzymes begin after the change in the differentiation profile of brain stem cells. In the process of spontaneous development of EAE and its accelerated development after treatment of mice, with MOG, DNA-met-BSA, and DNA-histones, the differentiation profiles are significantly different (Appendix A). Nevertheless, all these antigens ultimately lead to the accelerated development of EAE [9,10,11]. Interestingly, the diversity in the differentiation profiles of bone marrow stem cells during spontaneous development of EAE and after immunization of mice with three different antigens leads to the production of abzymes that hydrolyze MOG and DNA at different periods of EAE development. Wherein, the increase in the activity of all types of abzymes during spontaneous development of EAE is relatively gradual (Figure 3). The production of abzymes that hydrolyze MOG during immunization of mice with any of the three antigens occurs mainly from 7 to 20 days, corresponding to the onset and acute phase of EAE (Figure 3). A sharp increase in DNase activity of abzymes from 7 to 20 days occurs only when mice are treated with MOG (Figure 3b). Immunization of mice with antigens containing DNA (DNA-met-BSA and DNA-histones) leads to a delay in the production of abzymes hydrolyzing DNA. However, in these cases during the period corresponding to remission, this activity becomes 5–8-fold higher than the maximum activity after immunization of the mice with MOG (Figure 3). It is interesting that both during the spontaneous development of EAE (Figure 3a) and after immunization of mice with MOG or DNA-met-BSA, a sharp increase in peroxidase activity during the onset and acute phase of EAE is not observed (Figure 3b,c). Some repetition of the profile of changes in MOG-hydrolyzing activity with a maximum during the acute phase for peroxidase activity is observed only after mice immunization with a DNA-histones complex (Figure 3d).

In the case of many compounds, including proteins, DNAs, RNAs, oligosaccharides, lipids, etc.), a possible way of production antibodies and abzymes to these antigens is evident and well described [15,16,17,18,19,20]. The origin of abzymes with redox functions is not yet clear. It is possible to assume several ways of their origin theoretically. At first, abzymes with different redox activities in healthy humans and patients with AIDs can simply reflect the constitutive synthesis of germline Abs described by the Paul group [51,52]. The production of such abzymes, in addition, can probably be stimulated by various toxic, mutagenic, and carcinogenic compounds forming complexes with some proteins, which can lead to the production of abzymes against these haptens. The second antiidiotypic antibodies against various enzymes’ catalytic sites may also possess different catalytic activities [15,16,17,18,19,20]. Thus, in principle, one cannot exclude that different redox abzymes can be antiidiotypic antibodies against canonical catalases, peroxidases, glutathione peroxidases, superoxide dismutases, and other enzymes oxidizing different compounds.

It is believed that during MS development, at first, a normal immune response can be stimulated by various foreign antigens during different bacterial and viral infections [53,54,55,56,57]. Molecular mimicry due to homology between components of humans and bacterial-viral agents such as hepatitis B, Epstein-Barr, measles, influenza, herpes simplex, and papilloma viruses may be involved in the autoimmune MS pathogenesis [57]. Some antigens of some viruses (and/or bacteria) can penetrate through blood-brain barrier, change the differentiation profile of bone marrow stem cells, and stimulate the formation of B lymphocytes producing different abzymes, including ones with oxidation-reduction functions. Interestingly, the relative activity of IgGs from the cerebrospinal fluid of MS patients in the hydrolysis of MBP, DNA, and oligosaccharides, depending on the type of activity, is in average 30–60 times higher than from the blood serum of the same patients [58,59,60]. Consequently, the differentiation of cells leading to the formation of lymphocytes producing abzymes can begin already at the level of cerebrospinal fluid.

It should be emphasized that the blood of healthy people and animals usually does not contain abzymes, except for those that catalyze redox reactions. The emergence of abzymes is clearly associated with the development of various AIDs [15,16,17,18,19,20].

Considering the data discussed above, the question arises why antibodies with redox functions are needed. The predominance of the antioxidant system’s prooxidant processes and insufficiency can lead to abnormalities of myelination, hypofunction of NMDA receptors, death of interneurons, and other very different abnormal processes, and thereby contribute to the development of pathology [46,47]. As noted above, antibodies from healthy donors and animals, as well as from patients with AIDs, have not only peroxidase, but also H_2_O_2_-independent oxidoreductase, catalase, and superoxide dismutase activities [16,17,18,19,20,31,32,33,34,35,36,37,38,39].

The literature shows that mouse EAE has a complex neuropharmacology and many of the drugs that are in current or imminent use in MS have been developed, tested or validated on the basis of EAE studies [61,62,63]. This makes EAE a very versatile system to use in translational neuro- and immunopharmacology [63]. Environmental and microbial toxins, drugs, organic solvents and heavy metals can induce onset and progression of MS [61,62,63]. In EAE mice model these different harmful compounds (xenobiotics) can induce peripheral autoimmune conditions that trigger loss of tolerance, autoreactivity and ultimately autoimmune diseases [61,62,63].

It should be noted that abzymes with oxidoreductase activity from the blood of C57BL/6 mice, humans [37,38,39], and rats [33,35] oxidize not only the three substrates that we used in this work, but also many other compounds. From our point of view, some of the xenobiotics and drugs can also be good substrates for human and animal abzymes.

Our data on abzymes with redox functions suggest that catalytic antibodies oxidizing different compounds harmful to humans can reduce oxidative disorders’ pathological effects and protect the body from oxidative stress. Due to the high content in blood and prolonged circulation of IgGs with redox activities, they can decrease oxidative disorders in the bloodstream. In addition, the ability of antibodies to be accumulated in the foci of inflammation, catalytic IgG with peroxidase and catalase activities, along with canonical catalases and glutathione peroxidases [64], can participate in the regulation of H_2_O_2_ concentration and limiting the damage caused by a high concentration of hydrogen peroxide and oxidative processes in these zones.

The biological significance of abzymes with redox functions in humans and animals may be much important than it might seem at first glance. It was shown that polyclonal and monoclonal antibodies from various mammals have superoxide dismutase activity and efficiently reduce singlet oxygen (^1^O_2_^*^) to ^•^O_2_^−^ leading to H_2_O_3_ as the first intermediate, while a special cascade of chemical reactions finally resulting in the formation of H_2_O_2_ [65,66]. The authors supposed that the origin of this catalytic activity is based on adaptive maturation of the variable domains of Abs. These results demonstrate a possible protective function of these abzymes and raise the question of whether detoxification of ^1^O_2_^*^ can play an important role in the evolution of antibodies. These catalytic antibodies demonstrate possible mechanisms of how oxygen can be reduced and recycled in phagocyte action, thus increasing the anti-microbicidal action of the immune system [65,66]. Therefore, it seems likely that in healthy mammals exist a whole set of abzymes with different redox protective functions. Some Abs with superoxide dismutase activity can convert oxygen into hydrogen peroxide, while other Abzs with peroxidase and catalase activities to neutralize H_2_O_2_. Taking into account a set of many different dangerous for mammals compounds as substrates of peroxidase and oxidoreductase IgGs [31,32,33,34,35,36,37,38,39], it is easy to propose that a set of possible substrates of catalytic antibodies can be extremely wide including some drugs and xenobiotics. Taken together, polyclonal Abzs can serve as an additional system of human and various animals of reactive oxygen species detoxification.

## 4. Materials and Methods

### 4.1. Reagents

If not stated otherwise, chemicals, proteins, Protein G-Sepharose, Protein A-Sepharose, and the Superdex 200 HR 10/30 columns were purchased from (GE Healthcare Life Sciences, New York, NY, USA). 3,3′-Diaminobenzidine (DAB, D8001, Sigma-Aldrich, (St. Louis, MO, USA)); 2,2′-azino-bis(3-ethylbenzothiazoline-6-sulfonic acid) diammonium salt (ABTS, A1888, Sigma-Aldrich, St. Louis, MO, USA); *o*-phenylenediamine (OPD, P9029, Sigma-Aldrich, St. Louis, MO, USA). MOG_35–55_ was from EZBiolab (Nugemberg, Germany), *Bordetella pertussis* toxin (*M. tuberculosis)* from Native Antigen Company (Oxfordshire, UK). These preparations were free from lipids, oligosaccharides, nucleic acid, and other possible contaminants.

### 4.2. Experimental Animals

C57BL/6 mice (three months of age) were grown in the breeding facility for mice of the Institute of Cytology and Genetics (ICG) in standard conditions free of any bacterial, viral, and other pathogens. All experiments were performed by us in accordance with the Institute of Cytology and Genetics Bioethical committee’s protocols corresponding to humane principles of work with animals of the European Communities Council Directive 86/609/CEE. The Bioethical ICG committee supported our study [9,10,11].

### 4.3. Immunization of Mice

In this article were used IgG preparations used earlier for analysis of different parameters characterizing the development of EAE in C57BL/6 mice before and after their immunization with MOG [9], the complex of polymeric bovine DNA with methylated bovine serum albumin (DNA-metBSA) [10], and complex of DNA with five histones (H1, H2A, H2B, H3, and H4; DNA-histones) [11]. On the first day (zero time), mice were immunized with MOG, DNA-metBSA, or DNA-histones as described earlier [9,10,11]. For purification of IgGs of 0.7–1 mL of the blood was collected 7-80 days after immunization by decapitation using standard approaches as in [9,10,11]. Methods of immunization of mice have been published earlier [9,10,11], and their more detailed description is given in Appendix A (Part 1, Immunization of mice).

### 4.4. IgG Purification

Electrophoretically and immunologically homogeneous IgG preparations were obtained as in [9,10,11] by sequential chromatography of the blood plasma proteins on Protein G-Sepharose and following FPLC gel filtration in drastic conditions (pH 2.6). For the protection of IgGs from possible bacterial and viral contamination, they were filtered through Amicon filters units (0.1 μm). The resulting solutions were divided into aliquots and kept in sterilized tubes at −70 °C before using them in various experiments. SDS-PAGE analysis of IgGs for homogeneity was done in 4–15% gradient gels; the proteins were visualized by silver staining [9,10,11]. To exclude possible artifacts due to hypothetical traces of contaminating enzymes, IgGs were separated by SDS-PAGE. Their protease and nuclease activities were detected using a gel assay, as in [9,10,11]. These activities were only revealed in the band corresponding to intact IgGs, and there were no other peaks of proteins, protease, or DNase activities. Methods of IgG purification have been published earlier [9,10,11], and their more detailed description is given in Appendix A (Part 2, IgG purification).

### 4.5. Assay of Redox Activities

Estimation of relative oxidoreductase and peroxidase activities of IgG were performed using optimal conditions [37,38,39,40]. Reaction mixtures (100–150 µL) contain of 25 mM K-phosphate (pH 6.8), with or without 10 mM H_2_O_2_, one of different substrates (0.07–0.55 mM) and 0.1–0.5 mg/mL IgGs. Oxidation of ABTS, DAB, and OPD was detected from the optical density changes at 450 nM (A_450_) using 0.1 cm quartz cuvettes and Genesis 10S Bio spectrophotometer (Thermo Fisher Scientific, Inc.; Pittsburgh, PA, USA). Reaction mixtures were incubated at 23 °C, and time dependencies (0.5–20 min) of A_450_ change were analyzed. Reaction mixtures containing no IgGs were used as controls.

The *K*_M_ and *V*_max_ (*k*_cat_) values were calculated from the kinetic data by least-squares nonlinear regression fitting using Microcal Origin v5.0 software and presented as linear transformations using a Lineweaver-Burk plot according to [67]. Errors in the values were within 10–15%.

### 4.6. In Situ Analysis of Peroxidase Activity

To exclude possible artifacts due to hypothetical traces of contaminating enzymes, IgGs were separated by SDS-PAGE. Their protease and nuclease activities were detected using a gel assay, as in [9,10,11]. SDS-PAGE analysis of 14 μg IgGs homogeneity and in situ peroxidase activity was performed in 4–15% gradient gels (0.1% SDS) under non-reducing conditions [37,39]. After SDS-PAGE to restore the peroxidase activity, SDS was removed using the gel incubation for 1 h at 23 °C with K-phosphate (pH 6.8). The gel was washed ten times with this buffer. To allow IgGs refolding and to assay for peroxidase activity, several longitudinal slices of the gel were incubated in the reaction mixture containing 0.2 mg/mL DAB, 10 mM H_2_O_2_ and 1.0 mM CuCl_2_ for 10 h until a colored insoluble DAB oxidation product was formed. The parallel longitudinal lanes were used to detect the IgG position on the gel by silver staining.

### 4.7. Statistical Analysis

The results are reported as the mean and standard deviation of at least 2–3 independent experiments for each IgG preparation, averaged over seven different mice in every group.

## 5. Conclusions

Taken together, our results show that autoimmune-prone C57BL/6 mice are characterized by the spontaneous and different antigens accelerated development of EAE associated with changes in the differentiation of bone marrow HSCs and production of abzymes hydrolyzing MBP, MOG, and DNA. Such specific disorders of the immune status of mice during the development of EAE also lead to an increase in the activity of autoantibodies with peroxidase and oxidoreductase activities oxidizing different substrates of canonical redox enzymes. Thus, redox abzymes with increased activity in MS, SLE patients, and EAE prone mice can protect them from some harmful compounds somewhat better than in healthy peoples and animals.

## Figures and Tables

**Figure 1 molecules-26-02077-f001:**
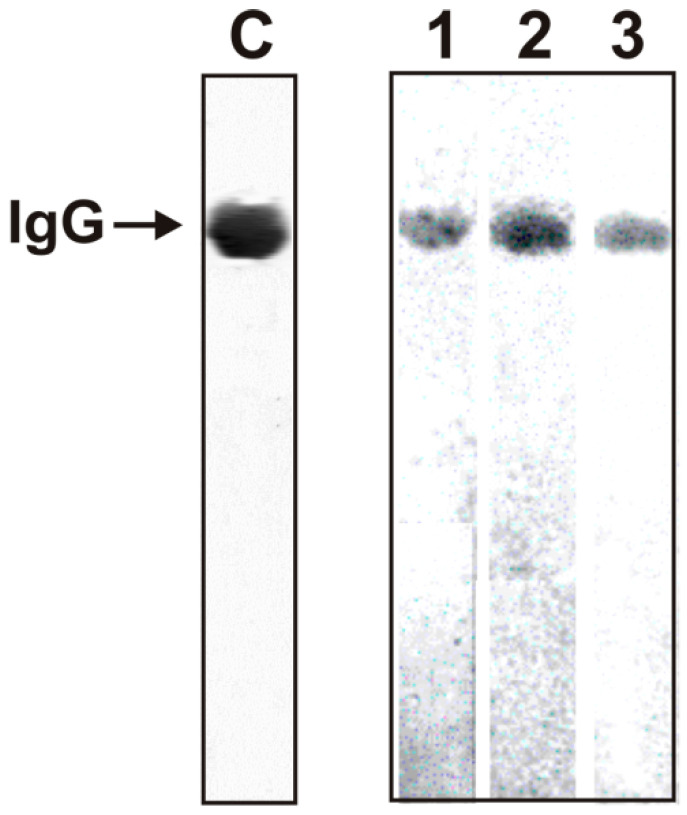
In situ SDS-PAGE analysis of 14 µM IgGs peroxidase activities using a nonreducing 4–15% gradient gel. Equimolar mixtures of 21 preparations corresponding IgGs from mice treated with different antigens: MOG (**lane 1**), DNA-metBSA (**lane 2**), and DNA-histones (**lane 3**). After the electrophoresis, the gels were incubated under special conditions for protein refolding. Then, IgGs’ peroxidase activity from different mice was revealed by incubating longitudinal gel slices in the reaction mixture containing DAB, H_2_O_2,_ and CuCl_2_. The gels were incubated for 10 h until a colored insoluble oxidation product was formed. Longitudinal slices of the gels were stained with silver (**lane C**) to reveal IgG positions.

**Figure 2 molecules-26-02077-f002:**
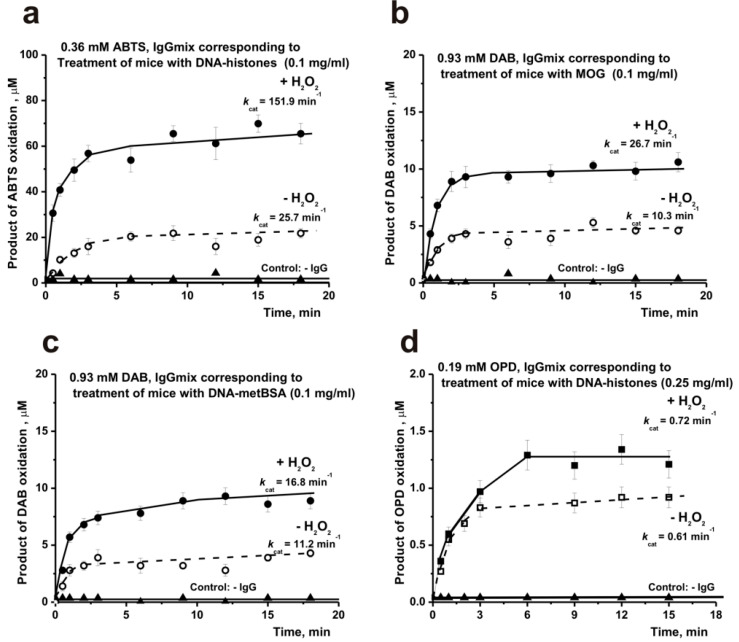
Typical examples of the time-dependences of products formation in the reaction of three substrates (ABTS (**a**), DAB (**b**,**c**), and OPD (**d**)) oxidation catalyzed by IgG_mix_ preparations in the presence and in the absence of H_2_O_2_. All designations are given on Panels **a**–**d**.

**Figure 3 molecules-26-02077-f003:**
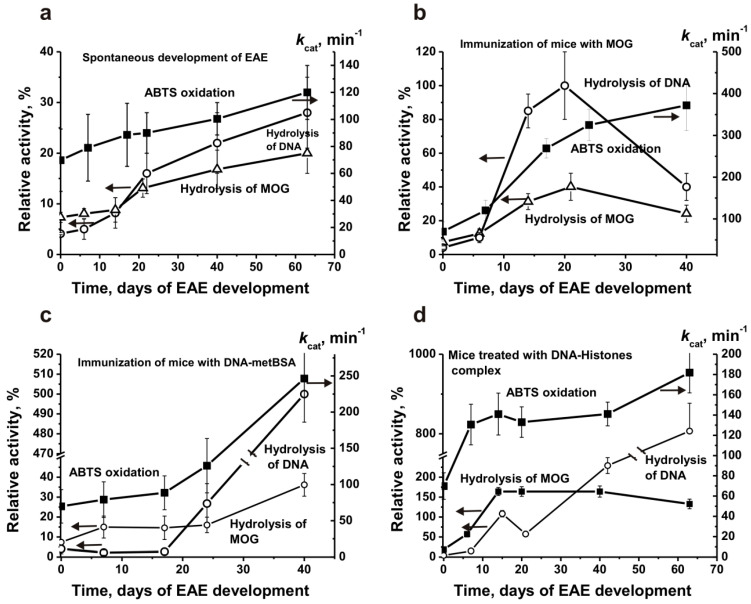
Over time changes in the relative activity (5 of the hydrolysis) of IgGs against DNA and MOG in the hydrolysis of these substrates (**a**–**d**; left scales) as well as oxidation of ABTS (*k*_cat_, min^−1^; **a**–**d**; right scales). The mean values of the activities of IgGs from seven mice during spontaneous (**a**) as well as accelerated development of EAE after immunization of mice with MOG (**b**), DNA-metBSA (**c**), and DNA-histones (**d**) are given. All designations are marked in the Panels: the arrows in the Panels indicate to which Y axis the given curve belongs to the right or to the left.

**Figure 4 molecules-26-02077-f004:**
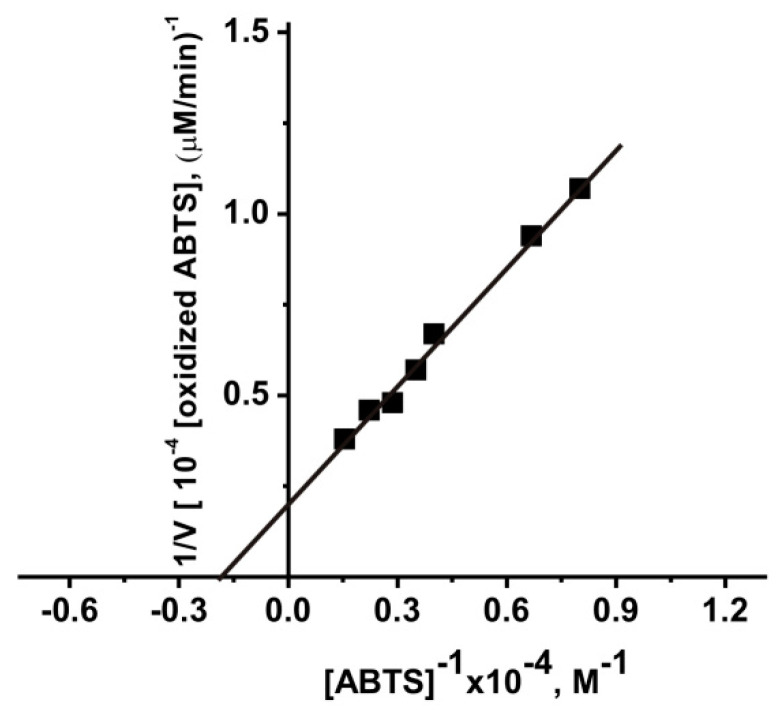
Determination of the *K*_m_ and *V*_max_ (*k*_cat_) values for ABTS in the reaction of its oxidation by one IgG preparation (0.15 mg/mL) using Lineweaver-Burk plots in the presence of H_2_O_2_. One IgG preparation with maximal activity corresponding to mice immunization with MOG was used.

## Data Availability

The data presented in this study are available on request from the corresponding author.

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
