# Peer review of "Increase in Autoantibodies-Abzymes with Peroxidase and Oxidoreductase Activities in Experimental Autoimmune Encephalomyelitis Mice during the Development of EAE Pathology"

_molecules, 2021, doi:10.3390/molecules26072077_

Round 1
Reviewer 1 Report
The authors aimed to determine the contribution of abzymes to the development of EAE.
This reviewer has several concerns:
- The authors need to describe the manuscript more accurately. For example, if the authors used MOG35-55 for immunization, the authors need to describe MOG35-55, not MOG.
- What DNA did the authors use for immunization?
- Although the authors described that C57BL/6 mice developed spontaneous EAE, is this true? Wildtype C57BL/6 mice rarely develop spontaneous EAE. Even if it is transgenic 2D2 mice that have MOG-specific T cell receptors (C57BL/6 background), the incidence is only 4% at 2.5-5 months old and 40% in the first year according to the Jackson Laboratory.
- The authors need to show the clinical courses of EAE mice immunized with MOG, DNA-metBSA, and DNA-histones.
- In Figure 2, the authors used mice with DNA-histone for ABTS and OPD, while the authors used mice with MOG for DAB. Why? How about other substrates or other treatments?
- The authors need to spell out several abbreviations: “Abs” on page 2, “CSF” on page 3, “BFU-E”, “CFU-E”, “CFUGM”, CFU-GEMM, OPD on page 4.
Author Response
- The authors need to describe the manuscript more accurately. For example, if the authors used MOG35-55 for immunization, the authors need to describe MOG35-55, not MOG.
Answer: It was corrected
- What DNA did the authors use for immunization?
Answer: It was corrected: polymeric bovine DNA
- Although the authors described that C57BL/6 mice developed spontaneous EAE, is this true? Wildtype C57BL/6 mice rarely develop spontaneous EAE. Even if it is transgenic 2D2 mice that have MOG-specific T cell receptors (C57BL/6 background), the incidence is only 4% at 2.5-5 months old and 40% in the first year according to the Jackson Laboratory.
Answer: Nevertheless, EAE develops very slowly in C57BL/6 mice, which in the early stages of disease is detected only by a change in the differentiation profile of bone marrow stem cells and the appearance of abzymes.
As it mentioned at the beginning of “Introduction” different autoimmune diseases (AIDs) were supposed first to be originated from hematopoietic stem cells (HSCs) defects [5]. The spontaneous and antigen-induced development of systemic lupus erythematosus (SLE) in MRL-lpr/lpr mice [6-8] and EAE in C57BL/6 mice [9-11] was later demonstrated is a consequence of the immune system-specific reorganization of bone marrow HSCs. The immune system defects in AIDs consist of specific changes in the profile of bone marrow HSCs differentiation combined with the production of specific catalytic abzymes-auto-Abs hydrolyzing DNAs, RNAs, polysaccharides, peptides, and proteins [6-11]. In addition, the auto-Abs with catalytic activities was revealed as statistically significant and the earliest markers of many autoimmune diseases development, which appear immediately after the start of a change in the differentiation profile of bone marrow stem cells [15-20]. Therefore, the early stages of development of AIDs can be detected only by changes in these parameters [9-20].
- The authors need to show the clinical courses of EAE mice immunized with MOG, DNA-metBSA, and DNA-histones.
Answer: Sorry, but clinical courses of EAE C57BL/6 mice immunized with MOG were described earlier in [3,4]. According to our earlier data, the appearance of clinical indicators after immunization of mice with MOG the same as in [3,4], while they are slightly different after immunization with DNA-metBSA and DNA-histones.
- In Figure 2, the authors used mice with DNA-histone for ABTS and OPD, while the authors used mice with MOG for DAB. Why? How about other substrates or other treatments?
Answer: As previously was shown, the optimal substrates for IgG antibodies with peroxidase activity from the blood of healthy rats [31-36], healthy donors, and patients with AIDs are ABTS, DAB, and OPD [37-39]. On the one hand, it does not really matter which substrate to use to assess the redox activity of abzymes from the blood of mice, they effectively oxidize all three substrates: ABTS, OPD, and DAB. It has been shown that all three substrates are efficiently oxidized by antibodies from the blood of mice immunized MOG, DNA-metBSA, and DNA-histones. To show that after immunization of mice with all immunogens, we have shown two substrates (ABTS and OPD) for abzymes after immunization of mice with DNA-histone, and DAB in the case of MOG. To show that oxidation of all three substrates occurs after immunization with any of the antigens in Figure 2, we added kinetic curves of DAB oxidation by antibodies after immunization of mice with DNA-metBSA. In addition, ABTS is the best substrate for mice IgGs with peroxidase activity, the oxidation of which occurs 5.7-9 and 211 times faster than DAB and OPD, respectively. Taking this into account, the relative activities of various IgG preparations corresponding to different stages of EAE development in mice were evaluated using ABTS as the main substrate.
The authors need to spell out several abbreviations: “Abs” on page 2, “CSF” on page 3, “BFU-E”, “CFU-E”, “CFUGM”, CFU-GEMM, OPD on page 4.
Answer: It was done
Thank you very much for your comments, which made it possible to make the article better.
Sincerely
Prof. Georgy A. Nevinsky
Reviewer 2 Report
The authors used a mouse model of MS (EAE C57BL/6 mice)to analyze changes in the activity of antibodies during the onset, acute phase and remission of this pathology The authors assessed the change in the relative redox activities of IgGs antibodies from the blood of C57BL/6 mice at different stages of experimental autoimmune encephalomyelitis (EAE) development. The peroxidase activity of mice IgGs was on average 6.9-fold higher than the oxidoreductase activity. The peroxidase activity of IgGs increased during the spontaneous development of EAE during 40 days, 1.4-fold. After EAE development acceleration due to mice immunization with MOG (5.3-fold), complexes of DNA with methylated bovine serum albumin (DNA-metBSA; 3.5-fold), or with histones (2.6-fold), the activity was increased much faster. The results show that IgGs' redox activities can play an important role in the protection of mice from toxic compounds and oxidative stress. The authors compared previously obtained data on specific overtime changes in the profile of bone marrow bone marrow hematopoietic stem cells (HSCs) differentiation and appearance in the blood of catalytic abzymes splitting DNA, MOG, and MBP with in time changes in IgGs peroxidase and oxidoreductase activities. The autoimmune prone C57BL/6 mice are characterized by the spontaneous and different antigens accelerated development of EAE associated with changes in the differentiation of bone marrow HSCs and production of abzymes hydrolyzing MBP, MOG, and DNA. The redox abzymes with increased activity in multiple sclerosis (MS), SLE patients and EAE prone mice can protect them from some harmful compounds somewhat better than in healthy peoples and animals.
Comments:
- The literature shows that EAE has a complex neuropharmacology and many of the drugs that are in current or imminent use in MS have been developed, tested or validated on the basis of EAE studies. Experiments indicate that there is great heterogeneity in the susceptibility to the induction, the method of induction and the response to various immunological or neuropharmacological interventions. This makes EAE a very versatile system to use in translational neuro- and immunopharmacology, but the model needs to be tailored to the scientific question being asked.
- Researchers indicate that difficulties and underscoring the inherent weaknesses of this model of MS in straightforward translation from EAE to the human disease. This variability explores multiple facets of the immune and neural mechanisms of immune-mediated neuroinflammation and demyelination as well as intrinsic protective mechanisms.
- In this model can harmful compounds (xenobiotics) induce peripheral autoimmune conditions that trigger loss of tolerance, autoreactivity and ultimately autoimmune disease that are relevant to antibodies in this mouse model of MS (EAE C57BL/6 mice) during the onset, acute phase and remission of this pathology. In MS patients xenobiotic compound may induce autoreactive T and B lymphocytes that lead to persistent autoimmunity.
- Can the redox abzymes and IgGs peroxidase and oxidoreductase activities with increased changes in the activity in multiple sclerosis (MS), SLE patients and EAE prone mice protect them from the systemic autoimmune disease (xenobiotic-induced) that may be found in this model and MS and SLE patients?
RELEVANT REFERENCES:
A. Hachim MY, Elemam NM, Maghazachi AA. The Beneficial and Debilitating Effects of Environmental and Microbial Toxins, Drugs, Organic Solvents and Heavy Metals on the Onset and Progression of Multiple Sclerosis. Toxins (Basel). 2019;11(3):147.
B. Michael Pollard,a Joseph M. Christy,a David M. Cauvi,b and Dwight H. Konoc. Environmental Xenobiotic Exposure and Autoimmunity. Curr Opin Toxicol. 2018 Aug; 10: 15–22.
C. Cris S Constantinescu, Nasr Farooqi, Kate O'Brien, and Bruno Gran. Experimental autoimmune encephalomyelitis (EAE) as a model for multiple sclerosis (MS). Br J Pharmacol. 2011 Oct; 164(4): 1079–1106.
Author Response
Second Referee
The authors used a mouse model of MS (EAE C57BL/6 mice)to analyze changes in the activity of antibodies during the onset, acute phase and remission of this pathology The authors assessed the change in the relative redox activities of IgGs antibodies from the blood of C57BL/6 mice at different stages of experimental autoimmune encephalomyelitis (EAE) development. The peroxidase activity of mice IgGs was on average 6.9-fold higher than the oxidoreductase activity. The peroxidase activity of IgGs increased during the spontaneous development of EAE during 40 days, 1.4-fold. After EAE development acceleration due to mice immunization with MOG (5.3-fold), complexes of DNA with methylated bovine serum albumin (DNA-metBSA; 3.5-fold), or with histones (2.6-fold), the activity was increased much faster. The results show that IgGs' redox activities can play an important role in the protection of mice from toxic compounds and oxidative stress. The authors compared previously obtained data on specific overtime changes in the profile of bone marrow bone marrow hematopoietic stem cells (HSCs) differentiation and appearance in the blood of catalytic abzymes splitting DNA, MOG, and MBP with in time changes in IgGs peroxidase and oxidoreductase activities. The autoimmune prone C57BL/6 mice are characterized by the spontaneous and different antigens accelerated development of EAE associated with changes in the differentiation of bone marrow HSCs and production of abzymes hydrolyzing MBP, MOG, and DNA. The redox abzymes with increased activity in multiple sclerosis (MS), SLE patients and EAE prone mice can protect them from some harmful compounds somewhat better than in healthy peoples and animals.
Comments:
- The literature shows that EAE has a complex neuropharmacology and many of the drugs that are in current or imminent use in MS have been developed, tested or validated on the basis of EAE studies. Experiments indicate that there is great heterogeneity in the susceptibility to the induction, the method of induction and the response to various immunological or neuropharmacological interventions. This makes EAE a very versatile system to use in translational neuro- and immunopharmacology, but the model needs to be tailored to the scientific question being asked.
- Researchers indicate that difficulties and underscoring the inherent weaknesses of this model of MS in straightforward translation from EAE to the human disease. This variability explores multiple facets of the immune and neural mechanisms of immune-mediated neuroinflammation and demyelination as well as intrinsic protective mechanisms.
- In this model can harmful compounds (xenobiotics) induce peripheral autoimmune conditions that trigger loss of tolerance, autoreactivity and ultimately autoimmune disease that are relevant to antibodies in this mouse model of MS (EAE C57BL/6 mice) during the onset, acute phase and remission of this pathology. In MS patients xenobiotic compound may induce autoreactive T and B lymphocytes that lead to persistent autoimmunity.
- Can the redox abzymes and IgGs peroxidase and oxidoreductase activities with increased changes in the activity in multiple sclerosis (MS), SLE patients and EAE prone mice protect them from the systemic autoimmune disease (xenobiotic-induced) that may be found in this model and MS and SLE patients?
RELEVANT REFERENCES:
- Hachim MY, Elemam NM, Maghazachi AA. The Beneficial and Debilitating Effects of Environmental and Microbial Toxins, Drugs, Organic Solvents and Heavy Metals on the Onset and Progression of Multiple Sclerosis. Toxins (Basel). 2019;11(3):147.
- Michael Pollard,a Joseph M. Christy,a David M. Cauvi,b and Dwight H. Konoc. Environmental Xenobiotic Exposure and Autoimmunity. Curr Opin Toxicol. 2018 Aug; 10: 15–22.
- Cris S Constantinescu, Nasr Farooqi, Kate O'Brien, and Bruno Gran. Experimental autoimmune encephalomyelitis (EAE) as a model for multiple sclerosis (MS). Br J Pharmacol. 2011 Oct; 164(4): 1079–1106.
Answer: Your very useful data and comments are of an unusual and somewhat philosophical character, thank you for them and for the very useful literary references. Taking your comments into account, we have added the following text to the discussion.
The literature shows that EAE has a complex neuropharmacology and many of the drugs that are in current or imminent use in MS have been developed, tested or validated on the basis of EAE studies [61-63]. This makes EAE a very versatile system to use in translational neuro- and immunopharmacology [63]. Environmental and microbial toxins, drugs, organic solvents and heavy metals can induce onset and progression of MS [61-63]. In EAE mice model these different harmful compounds (xenobiotics) can induce peripheral autoimmune conditions that trigger loss of tolerance, autoreactivity and ultimately autoimmune diseases [61-63].
It should be noted that abzymes with oxidoreductase activity from the blood of C57BL/6 mice, humans [37-39], and rats [33,35] oxidize not only the three substrates that we used in this work, but also many other compounds. From our point of view, some of the xenobiotics and drugs can also be good substrates for human and animal abzymes.
And later:
Taking into account a set of many different dangerous for mammals compounds as substrates of peroxidase and oxidoreductase IgGs [31-39], it is easy to propose that a set of possible substrates of catalytic antibodies can be extremely wide including some drugs and xenobiotics.
Thank you very much for your comments and references, which made it possible to make the article better.
Sincerely
Prof. Georgy A. Nevinsky
Round 2
Reviewer 1 Report
The authors responded to my concerns. However, they are not enough. Although the MOG-EAE model is widely used as a mouse model for multiple sclerosis, the other mice (spontaneous EAE of wildtype C57BL/6 mice and mice immunized with DNA-metBSA and DNA-histones) seems to be used only in the articles from the authors. In their previous articles, although the authors cited an article by Croxford AL et al for the explanation of EAE models, there is no description of spontaneous EAE of wildtype C57BL/6 mice. The authors did not show critical data, such as clinical course, to estimate the validity of these mice as EAE models.
Since these mice may be novel useful models for multiple sclerosis studies, this reviewer requests the authors to include the data of clinical courses and histological staining of the brains of these mice.
Reviewer 2 Report
Thank you for the effort and expertise with relevance to the revised manuscript and to ensure that the research is properly verified without any inadvertent errors.